# Automated Rice Seedling Segmentation and Unsupervised Health Assessment Using Segment Anything Model with Multi-Modal Feature Analysis

**DOI:** 10.3390/s25175546

**Published:** 2025-09-05

**Authors:** Hassan Rezvan, Mohammad Javad Valadan Zoej, Fahimeh Youssefi, Ebrahim Ghaderpour

**Affiliations:** 1Department of Photogrammetry and Remote Sensing, K. N. Toosi University of Technology, Tehran 19967-15433, Iran; h.rezvan@email.kntu.ac.ir (H.R.); or youssefi@usx.edu.cn (F.Y.); 2Institute of Artificial Intelligence, USX, Shaoxing University, 508 West Huancheng Road, Yuecheng District, Shaoxing 312000, China; 3Department of Earth Sciences & CERI Research Centre, Sapienza University of Rome, P.le Aldo Moro, 5, 00185 Rome, Italy

**Keywords:** smart agriculture, crop growth monitoring, food security, remote sensing, feature fusion, deep learning, segment anything model

## Abstract

This research presents a fully automated two-step method for segmenting rice seedlings and assessing their health by integrating spectral, morphological, and textural features. Driven by the global need for increased food production, the proposed method enhances monitoring and control in agricultural processes. Seedling locations are first identified by the excess green minus excess red index, which enables automated point-prompt inputs for the segment anything model to achieve precise segmentation and masking. Morphological features are extracted from the generated masks, while spectral and textural features are derived from corresponding red–green–blue imagery. Health assessment is conducted through anomaly detection using a one-class support vector machine, which identifies seedlings exhibiting abnormal morphology or spectral signatures suggesting stress. The proposed method is validated by visual inspection and Silhouette score, confirming effective separation of anomalies. For segmentation, the proposed method achieved mean dice scores ranging from 72.6 to 94.7. For plant health assessment, silhouette scores ranged from 0.31 to 0.44 across both datasets and various growth stages. Applied across three consecutive rice growth stages, the framework facilitates temporal monitoring of seedling health. The findings highlight the potential of advanced segmentation and anomaly detection techniques to support timely interventions, such as pruning or replacing unhealthy seedlings, to optimize crop yield.

## 1. Introduction

The global population is projected to approach 10 billion in the coming decades, intensifying the demand for food production and placing significant pressure on the agricultural sector to enhance productivity and sustainability [1,2,3,4,5]. Despite progress in reducing hunger and food insecurity worldwide, these efforts remain uneven, with an estimated 582 million people likely to face chronic undernourishment by 2030 if current trends continue [1]. Rice, a staple food for over half of the world’s population, especially in Asia, plays a critical role in global food security and poverty alleviation efforts [3,6,7,8,9,10].

To address the growing demand for food, smart farming and precision agriculture have emerged as essential approaches for enhancing rice production efficiency [4,9]. These technologies enable precise monitoring of crop health, identification of environmental stressors, quantification of plant stress, seedling counting, and early detection of defective plants [3,7,8,9,10,11,12]. Data-driven insights from these approaches support optimized decisions on planting density, irrigation, and resource management, ultimately contributing to higher yields and more sustainable practices [3,9,10].

Crop and seedling monitoring typically aims to (i) detect and localize individual plants and (ii) assess plant health or stress levels. Remote sensing, including optical and radar sensors, has been widely used for crop mapping [13,14]. However, unmanned aerial vehicle (UAV) imagery offers superior utility due to its higher spatial resolution for these tasks [12,14,15,16]. Deep learning and object detection models, such as fully convolutional networks, visual geometry group (VGG-16), scalable and efficient object detection (EfficientDet), MobileNetV2, Faster region-based convolutional neural network (R-CNN), and you only look once (YOLO) variants have been extensively applied for this rice seedling detection [3,5,7,8,10,17]. Similar methods have also been used for other crops such as sorghum and maize [15,16].

Beyond detection, health and stress assessment are essential for mitigating yield losses and ensuring crop quality [4,9,18,19]. Early detection of stress symptoms enables timely interventions to optimize growth conditions and resource use [4,9]. Various optical technologies, including red–green–blue (RGB), multispectral and hyperspectral imaging, thermography and 3D scanning, have been applied to detect plant diseases and stress factors such as nutrient deficiencies, drought and pathogens [11,18,20]. Structural attributes (e.g., plant height, leaf area, canopy structure) [20,21], spectral indices (e.g., normalized difference vegetation index, soil adjusted vegetation index, excess green index) [22,23,24,25,26,27,28], and textural features further enrich health assessments [11,19,29]. Comprehensive approaches that integrate these diverse features improve detection accuracy and reliability.

The segment anything model (SAM), a recent innovation in image segmentation, offers zero-shot generalization to segment unfamiliar objects without additional training [30,31,32]. SAM has been applied across diverse domains, including public health, habitat detection [32], medical imaging [33,34], and autonomous robotics [35]. SAM operates effectively with minimal human input, such as bounding boxes, points, or text-based prompts [32]. In agriculture, however, most health assessment tasks still depend on large, labeled datasets, which are costly, time-consuming to generate, and often lack scalability across diverse and dynamic agricultural landscapes [36].

Unsupervised techniques, such as clustering and anomaly detection, provide viable alternatives by enabling data analysis without labeled samples. Approaches like fuzzy c-means clustering, one-class support vector machine (OCSVM), and Isolation Forest have been used for tasks ranging from crop type mapping to stress detection and sensor fault identification in smart farming systems [37,38,39,40,41,42,43]. These methods are particularly valuable in agricultural contexts where labeled data are scarce.

Despite advancements in remote sensing and computer vision, significant gaps persist in rice seedling monitoring. Current methods often rely on complex, black-box deep learning architectures with limited focus on single growth stages and lack integration between segmentation outputs and actionable agronomic insights. These limitations hinder practical deployment in real-world agricultural scenarios.

To address these challenges, this study introduces a novel, label-free framework for automated rice seedling segmentation and stress assessment across three growth stages using UAV-acquired RGB imagery. The method integrates SAM for segmentation, extracts morphological, spectral, and textural features, and employs OCSVM for anomaly detection to identify potentially stressed or unhealthy seedlings. This study builds upon the authors’ previous work [44], where the applicability of the SAM was evaluated for rice seedling segmentation under three prompting scenarios: point prompts, bounding-box prompts, and automatic mask generation, all relying on user-provided inputs from the source dataset. In contrast, the present manuscript introduces a fully automated framework that eliminates the need for manual inputs. Additionally, this work advances beyond segmentation by analyzing seedling health. To check the scalability of the approach, the entire process was also applied to a second, independently curated dataset.

The key contributions of this study include

Proposing a lightweight, training-free framework for automated seedling segmentation.Developing an interpretable model combining spectral and morphological, and textural features.Implementing a stage-wise monitoring approach to capture temporal dynamics in seedling health.Bridging AI models with practical agricultural applications by linking segmentation outputs to field-based actions.

The remainder of this paper is organized as follows: Section 2 details the datasets, methodology, including segmentation, feature extraction, and health/stress assessment. Section 3 presents the results and evaluation. Section 4 discusses the implications and limitations, followed by conclusions in Section 5.

## 2. Materials and Methods

### 2.1. Study Regions and Rice Seedling Datasets

This study was conducted using two geographically and climatically distinct rice cultivation regions. The first study area is located in Wufeng District, Taichung, Taiwan, where UAV-based field surveys were carried out by the Department of Civil Engineering and the Innovation and Development Center of Sustainable Agriculture at National Chung Hsing University, in collaboration with the Taiwan Agricultural Research Institute (TARI) [7,10,45]. The images were taken using two sensors, DJI Phantom 4 Pro and DJI Zenmuse X7. The second study area is situated in Heilongjiang Province, in the northeastern region of China, characterized by lower annual temperatures and a high sensitivity of rice seedlings to sunlight and rainfall fluctuations [46]. These differences in environmental conditions provide diverse settings for validating the robustness of the proposed segmentation and monitoring frameworks. The location and elevation maps of these regions are illustrated in Figure 1.

Two benchmark datasets were used for model development and evaluation. The first dataset, collected in Taiwan, comprises 600 high-resolution RGB images acquired via a multi-rotor UAV that followed a pre-defined scouting route. UAV acquisitions were scheduled at fixed calendar dates to represent distinct growth stages, a practical compromise in the absence of detailed phenological measurements. Therefore, data were gathered across three distinct rice seedling growth stages: (1) early seedling stage (7 August 2018), featuring sparse, small seedlings; (2) mid-growth stage (14 August 2018), marked by denser canopy formation; and (3) mature stage (23 August 2018), characterized by full canopy coverage. The dataset includes both close-up images of individual seedlings and orthophotos of entire fields, supporting a range of tasks from detection to classification.

The second dataset, from China, was collected between 9 May and 16 June 2022, using RGB cameras installed at 11 meteorological stations. Images were captured seven times daily under consistent natural light conditions, using vertically mounted cameras 2.4 m above ground with a 90° field of view. Each image covers an area of 4.4 × 2.5 m. While this dataset includes both RGB and near-infrared (NIR) spectral data, only RGB images were utilized in this study [46]. One image per day was selected based on consistent lighting and time of capture, and the 17-day collection was grouped into three growth stages for analysis, while phenology data is not available. A summary of the information about the datasets is shown in Table 1.

Sample images of the different growth stages of rice seedlings used in this study are presented in Figure 2. Notably, in the second dataset, the rice field is visibly flooded during the initial growth stage.

### 2.2. Methodology

This section outlines the proposed methodology for rice seedling detection and unsupervised health/abnormality assessment. The framework consists of three main steps: seedling segmentation, feature extraction, and anomaly detection, as shown in Figure 3.

First, seedling locations are extracted using excess green minus excess red (ExGR) index, which enhances vegetation contrast in RGB images. Based on the extracted seedling positions, bounding boxes are automatically drawn to localize each seedling. These bounding boxes serve as point prompts for the SAM, which performs precise segmentation and masking of individual seedlings without requiring labeled data.

Following segmentation, a comprehensive set of features is extracted:Morphological features are derived from the generated masks to capture shape and structural characteristics.Spectral features are computed from the corresponding RGB imagery to assess color and reflectance properties.Textural features are extracted to quantify surface patterns and fine-grained variations that may indicate stress or disease.

These combined features serve as input to an OCSVM, an unsupervised anomaly detection algorithm. The OCSVM identifies seedlings with morphological or spectral deviations from the norm, flagging them as potentially stressed, unhealthy, or abnormal—without relying on pre-labeled datasets.

This integrated, label-free approach enables automated seedling monitoring and health assessment across multiple growth stages, facilitating early detection of issues for precision agriculture interventions.

#### 2.2.1. Automatic SAM for Rice Seedling Detection

To segment rice seedlings from UAV-acquired RGB imagery, a hybrid approach is implemented, which combines vegetation index enhancement with prompt-based segmentation. Initially, ExGR index in Equation (1) was computed to emphasize green regions corresponding to seedlings. Multi-Otsu thresholding and connected component analysis were then applied to automatically delineate bounding boxes around individual seedlings. The geometric center of each bounding box was extracted as a hint point to perform SAM, capable of producing high-quality instance masks without additional training. This workflow was applied across three growth stages to accommodate varying seedling morphologies with minimal manual input. The final output for each image was a binary mask highlighting segmented seedlings. Table 2 shows the SAM parameters.(1)ExGR=3Green−2.4Red−Blue

Segmentation performance was assessed using standard metrics, including mean Dice (mDice) and mean Intersection over Union (mIoU), and mean False Positive Rate (mFPR), alongside seedling count accuracy as a practical, agriculture-relevant indicator. The formulas of these metrics are as follows.(2)$$mDice=Mean\left(\frac{2\times TP}{2\times TP+FP+FN}\right)$$(3)$$mIoU=Mean\left(\frac{TP}{TP+FP+FN}\right)$$(4)$$mFPR=Mean\left(\frac{FP}{FP+TN}\right)$$
where TPs (True Positives) are pixels correctly identified as seedlings, while TNs (True Negatives) are pixels correctly identified as background. FPs (False Positives) occur when background pixels are incorrectly labeled as seedlings, and FNs (False Negatives) occur when actual seedling pixels are missed and labeled as background. Algorithm 1 shows this step. The visual diagram of Algorithm 1 is illustrated in Figure 4.
**Algorithm 1.** Automated Rice Seedling Detection1: Input: RGB image I of a rice seedling patch2: Output: Binary mask M with segmented seedlings3: Compute ExGR index from I to enhance green regions4: Apply multi-Otsu thresholding on ExGR to generate a binary map B5: **for each** connected component C in B **do**:5:            Compute bounding box BB_C around component C6:            Extract center point P_C of BB_C7:            Append P_C to the prompt list L8: Apply the SAM to I using point prompts in L9: Obtain segmentation mask M from SAM10: **Return** M

#### 2.2.2. Morphological-Spectral-Textural OCSVM-Based Anomaly Detection

From each segmented seedling mask, a set of morphologic, spectral, and textural features was extracted. Geometric features included Area, Perimeter, Solidity, Eccentricity, and Circularity, while spectral features comprised the intensities of the Red, Green, and Blue channels, along with the excess green (ExG) index. Moreover, contrast, dissimilarity, homogeneity, energy, correlation, and second moment computed from the gray-level co-occurrence matrix (GLCM) are represented as textural features. In the present research, textural features are extracted from GLCM using Scikit-image in Python 3.11. These extracted features collectively served as the input variables for the clustering analysis and subsequent anomaly detection processes. The formulas to calculate solidity, eccentricity, circularity and ExG index are as follows.

Solidity represents the ratio of a seedling’s area to its convex hull, reflecting shape compactness, where lower values may signal gaps or structural weakness. Eccentricity quantifies elongation, with abnormal values often linked to irregular growth. Circularity measures similarity to a circle and is highly sensitive to boundary irregularities such as notches or serrations. ExG emphasizes vegetation, with low values suggesting potential stress or poor health.(5)$$Solidity=\frac{Area}{Convex\_area}$$(6)$$Eccentricity=\sqrt{1-{\left(\frac{Minor\_axis}{Major\_axis}\right)}^{2}}$$(7)$$Circularity=\frac{4\pi \times Area}{Perimeter}$$(8)ExG=2Green−Red−Blue
where Area is the number of pixels inside the seedling mask, Convex_area is the smallest convex polygon that encloses the seedling mask, Perimeter is the length of the boundary of the seedling mask, and Major_ and Minor_axis lengths are the lengths of the longest and shortest axes of the best-fitting ellipse to the seedling shape.

An OCSVM was employed, trained solely on the extracted feature vectors under the assumption that most seedlings belonged to the healthy or normal class. OCSVM was chosen for its capability to detect anomalies within high-dimensional feature spaces while requiring minimal prior assumptions—particularly advantageous given the absence of labeled training data. A grid search optimization was performed to determine the best kernel type (polynomial, radial basis function, linear) and tune key hyperparameters, ν and γ. We utilized the silhouette score during the grid search to guide parameter selection and reduce overfitting, providing a robust unsupervised alternative in the absence of ground-truth labels. The final model classified each seedling as either an inlier (+1) or outlier (−1), with outliers interpreted as potentially stressed, damaged, or unhealthy seedlings.

To assess the performance of the proposed approach, several validation strategies were applied. First, anomaly seedlings were overlaid on UAV imagery to visualize the spatial distribution of detected outliers, allowing direct inspection of their physical characteristics. This qualitative step was carried out by the research team to verify whether detected anomalies corresponded to visibly distinct seedlings. Additionally, the Bhattacharyya distance was calculated to quantify the degree of overlap between clusters, with higher values indicating better separability [47]. The silhouette score, a widely used clustering metric [48,49], was also employed to evaluate the overall compactness and separation between inlier and outlier groups, where values approaching 1 reflect well-defined cluster boundaries. The process of this detection is shown in Algorithm 2:
**Algorithm 2.** Morphologic–Spectral–Textural OCSVM-Based Anomaly Detection1: Inputs: Set of segmented seedling masks, Corresponding RGB images2: Output: Outlier labels *O* (−1 for anomaly, +1 for normal) for each seedling3: Extract morphologic features (Area, Perimeter, Solidity, Eccentricity, Circularity)4: Extract spectral features (Red, Green, Blue intensities and ExG)5: Extract textural features (Contrast, Dissimilarity, Homogeneity, Energy, Correlation, 6: Second Moment)7: Form combined feature vectors F = {*f*_1_, *f*_2_, …, *f_n_}* ∈ ℝ*^d^*8: Perform grid search to optimize OCSVM parameters:9:                Kernel ∈ {linear, radial basis function, polynomial}10:                *ν* (nu) ∈ [0.01, 0.05, 0.1]11:                *γ* (gamma) ∈ [0.01, 0.1, auto, scale]12: Train OCSVM on F using best (kernel, *ν*, *γ*)13: Predict anomaly labels L ∈ {+1, −1} for each seedling14: Visual Validation and statistical metrics15: Return *O*

## 3. Results

This section presents the outcomes of the proposed framework, including the performance of the automatic segmentation using the SAM and the results of the unsupervised health and abnormality assessment of seedlings. Supporting analyses and validation metrics are also provided.

### 3.1. The Results of Automatic SAM for Rice Seedling Detection

#### 3.1.1. Segmentation Results

The segmentation results generated by the proposed method across the three rice growth stages are visually presented in Figure 5.

In addition to visual interpretation, the evaluation results using metrics are summarized in Table 3.

#### 3.1.2. Rice Seedling Counting

Counting seedlings poses a significant challenge in precision and smart agriculture. As part of the analysis, in addition to conventional segmentation performance metrics, seedlings identified by the proposed method were counted across all three growth stages to evaluate segmentation accuracy. The counted results were then compared with the actual number of seedlings using linear regression analysis, and the coefficient of determination (R^2^) was calculated to measure agreement. The resulting regression plots are shown in Figure 6.

### 3.2. Morphological-Spectral-Textural OCSVM-Based Anomaly Detection

After performing OCSVM, t-SNE is used to project features into 2D and colored points by OCSVM labels to visualize separation. The scatter plots are shown in Figure 7.

Also, plots of OCSVM decision function show how far points are from the decision boundary. These plots for each growth stage are illustrated in Figure 8.

For analyzing cluster-wise feature distribution, the Bhattacharyya distance calculated for each feature during the three growth stages is summarized in Table 4.

Finally, for validation, outliers in UAV images are visually inspected and evaluated. Some of the samples are shown in Figure 9.

Also, the results of evaluating the quality of clusters using silhouette score are summarized in Table 5.

## 4. Discussion

The SAM has been widely adopted across various fields for its flexible, prompt-based segmentation capabilities. As previously discussed, SAM typically relies on minimal human input. In this study, a fully automated seedling detection pipeline is developed that extracts representative hint points algorithmically, converting the segmentation task into a point-prompt framework without manual intervention. Both statistical evaluations and visual assessments confirmed the effectiveness of this approach. As illustrated in Figure 5, the proposed method successfully delineated rice seedlings and accurately segmented their boundaries across the early and mid-growth stages in both datasets. Although segmentation performance declined somewhat in the third stage due to dense canopy cover and overlapping seedlings along planting lines (the upper and lower boundaries in the first dataset and the right and left boundaries in second dataset) remained clearly distinguishable from the background, demonstrating the robustness and practical utility of the proposed framework even in challenging conditions.

In addition to visual evaluations, a comprehensive quantitative assessment is done using established segmentation metrics, including mIoU, mDice, and mFPR, to account for both segmentation accuracy and error rates (Table 3). Consistent with the visual observations, these metrics demonstrated higher performance for the first and second growth stages, with notably lower false positive rates and stronger boundary delineation. Performance degradation was observed in the third growth period, where increased canopy density and overlapping seedlings led to a higher rate of false positives and reduced segmentation accuracy. Beyond pixel-based metrics, seedling counting was also employed as a practical indicator of the model’s capability to detect and separate individual seedlings. As anticipated, the predicted seedling counts aligned closely with ground truth values in the first and second stages, reflected by higher R^2^ values from the regression analyses. However, this agreement diminished considerably in the third stage, as the merging of closely positioned seedlings along cultivated lines resulted in underestimation and a drop in the regression performance. These findings highlight both the strengths and current limitations of the proposed method in varying phenological conditions.

The selection of spectral, morphological, and textural features in this study was grounded in both agronomic relevance and their sensitivity to phenotypic variations associated with plant health. Mean values of the Red, Green, and Blue channels were included as healthy rice seedlings typically exhibit higher green reflectance, while stressed or diseased plants tend to display yellowing (low green, high red) or browning (low green with high red and blue). The ExG index further enhances this distinction, as lower ExG values often correspond to poor plant health. Morphological features were chosen to capture geometrical cues linked to plant health and structure. The area of each seedling mask reflects overall size, with smaller areas potentially indicating stunted or underdeveloped seedlings. Perimeter values provide insight into shape regularity, where healthy seedlings generally maintain smooth, compact contours, whereas stressed or abnormal plants often present with irregular, fragmented edges leading to larger perimeters (e.g., [50]). Eccentricity was included in detecting elongation or asymmetry arising from abnormal growth, while circularity measures how closely a shape approximates a circle, making it sensitive to serrations or notches at the seedling boundary. Solidity assesses structural integrity, with lower values suggesting gaps, deformities, or incomplete canopy cover. As shown in Table 4, these selected features proved relatively effective in separating normal seedlings from abnormal ones. Notably, circularity emerged as a strong discriminator during the second and third growth stages, reflected by favorable Bhattacharyya distance values. In contrast, during the first growth stage, textural attributes such as contrast and dissimilarity, together with some spectral features (e.g., Red and Green channels), exhibited higher discriminative power, reflecting the stronger color and texture differences between seedlings and background at early stages. Interestingly, in the third stage of the first dataset, the second moment also reached its maximum discriminative value, further supporting the stage-dependent feature shift. These results confirm the agronomic relevance and practical utility of the proposed feature set in monitoring seedling health.

The performance of the OCSVM anomaly detection model was assessed across both datasets, demonstrating its suitability for unsupervised monitoring in imbalanced seedling populations. As shown in Table 5 and Figure 8, the first dataset achieved Silhouette scores ranging from 0.44 to 0.41, with concentrated and well-separated score distributions in the early and mid-growth stages, indicating stable crop conditions and effective anomaly separation. Although a slight decrease in the Silhouette score was observed in the final stage, likely due to increased canopy overlap and stress heterogeneity, the anomalies remained distinctly separated from the normal seedlings near the decision boundary. In the second dataset, Silhouette scores were slightly lower, ranging from 0.34 to 0.31. The relatively lower scores can be attributed to more heterogeneous backgrounds and potential differences in UAV imaging distance, which collectively reduced the separability of feature distributions. Despite these lower values, the silhouette scores remain within an acceptable range for unsupervised clustering tasks. The anomaly score distributions initially exhibited moderate separation but became progressively wider and more dispersed in the later stages, reflecting heightened physiological variability under fluctuating environmental conditions. Despite these challenges, the OCSVM model, by the selected spectral, geometric, and textural features, consistently provided a reliable and moderately effective framework for temporal anomaly detection across both datasets without the need for labeled training data.

Since the approach was unsupervised, excluding features without ground-truth labels risks discarding potentially relevant anomaly-related information; therefore, all spectral and geometric features were retained to capture complementary variability. In addition, the silhouette scores remained stable when testing different feature subsets, indicating that the model performance was not overly sensitive to potential feature redundancy.

The t-SNE embeddings (Figure 7) provided a visual representation of the OCSVM’s ability to distinguish healthy seedlings from anomalies across growth stages in both datasets. In the first dataset, early-stage embeddings showed a few outliers positioned at the edges of dense clusters, reflecting predominantly healthy seedlings with minimal stress variation. Outliers increased and clustered in localized regions during the mid-growth stage, suggesting the emergence of homogeneous stress patterns. By the final stage, although outliers remained relatively sparse, they were more widely scattered as feature variability increased with canopy development. A similar trend was observed in the second dataset, where initial outliers appeared along cluster boundaries, followed by a gradual rise in scattered anomalies through the second and third stages, indicating growing physiological variability under fluctuating field conditions. These patterns align with expected plant stress dynamics as growth progresses.

According to the visual assessment of the samples of seedlings identified as unhealthy and stressed, and as illustrated in Figure 9, these seedlings often appear yellow or brown at different growth stages or exhibit notably smaller sizes at maturity compared to their healthy counterparts. This observation aligns with the findings in Table 4, where spectral and area-based morphological features show the greatest discriminatory power. Moreover, the temporal consistency in the health status of detected seedlings—with no sign of fluctuating classification across stages—suggests that the model captures persistent stress conditions rather than noise. The increasing number of identified anomalies across the three growth periods may therefore reflect actual physiological stress, growth variation, or responses to environmental factors.

The proposed framework demonstrates strong potential for scalability and practical deployment in agricultural monitoring. A key advantage is its unsupervised design, which does not rely heavily on annotated datasets that are often costly and labor-intensive to generate. By employing OCSVM for anomaly detection and the zero-shot capability of SAM for segmentation, the approach can be applied in a patch-wise manner, allowing efficient processing of large UAV datasets and enabling extension to broader spatial scales. Importantly, this zero-shot strategy preserves scalability, whereas fine-tuning SAM on task-specific data could improve performance but at the cost of reduced generalizability and higher dependency on annotated training datasets. At the same time, real-world challenges such as illumination variability, soil background heterogeneity, and crop phenological differences can influence segmentation and anomaly detection performance. To partially address these factors, the methodology was tested on two UAV datasets with distinct background conditions (bare soil and flooded fields) and under different illumination scenarios, demonstrating its robustness across variable environments. These results suggest that the framework is not only adaptable but also holds promise for wider application in diverse field conditions.

A limitation of this study is that SAM was applied with its default ViT-H backbone and point prompts, without model-level fine-tuning or domain-specific architectural adaptations to test the generalization capacity of foundation models. While this baseline implementation demonstrates the feasibility of SAM in rice seedling detection, future work will explore optimized prompting strategies and task-specific adaptations (e.g., fine-tuning) to improve robustness under complex conditions such as canopy overlap. While the proposed geometric-spectral-textural anomaly detection method using OCSVM effectively identifies stressed or unhealthy seedlings without requiring labeled data, several limitations must be acknowledged. First, the model is sensitive to the choice of hyperparameters such as the outlier fraction (ν) and to feature scaling, which can influence detection accuracy. Second, the absence of ground truth labels or field-based validation data limits the ability to confirm whether detected anomalies truly reflect biological or environmental stress, highlighting the need for future integration with in-field observations. Finally, due to the unsupervised nature of the model, there is a risk of misclassifying rare-but-healthy seedlings as anomalies.

## 5. Conclusions

As the global population continues to grow and food demand rises, the need for efficient and scalable agricultural monitoring has become increasingly critical to maximize productivity and ensure food security. Traditional field monitoring, reliant on manual inspection, remains labor-intensive, time-consuming and costly. In contrast, the integration of remote sensing technologies with automated, data-driven approaches offers a practical and scalable solution for crop monitoring throughout the growing season. In the present study, a two-stage unsupervised framework for rice seedling detection and health assessment is proposed, leveraging both geometric and spectral features. The first stage automated SAM using point-prompt to achieve precise segmentation of individual rice seedlings from RGB imagery. In the second stage, an OCSVM model was applied to a comprehensive set of extracted morphological, spectral and textural features to identify anomalous seedlings—potentially representing those under stress, damaged, or otherwise unhealthy. The integration of morphological features with spectral and textural indicators allowed for a comprehensive characterization of each seedling. The OCSVM model effectively detected anomalies, validated through visual inspection and silhouette scores, demonstrating robust performance in unsupervised seedling health assessment. The framework is fully automated, as both seedling extraction and health assessment are handled entirely through algorithmic procedures without human input. Specifically, ExGR thresholding was automatically determined using multi-Otsu, while OCSVM hyperparameters were optimized via an automated grid search, ensuring that the pipeline runs without any need for manual parameter adjustment. Furthermore, the application of this method across three distinct rice growth stages provided valuable insights into temporal patterns of plant development and stress responses, emphasizing the dynamic nature of crop health.

Key contributions of this work include

The automation of SAM for efficient seedling segmentation without manual annotations.The integration of multi-modal features (morphological, spectral, and textural) for comprehensive anomaly detection.The application of unsupervised learning for time-resolved monitoring of crop health in a scalable and interpretable framework.

These findings underscore the potential of advanced image segmentation and machine learning methods to support precision agriculture by enabling early detection of plant stress and guiding timely interventions to optimize crop yields.

Future research can build on this framework by incorporating more modalities, such as NIR and red-edge, thermal bands, 3D point clouds or chlorophyll fluorescence to extract physiological and structural indicators, such as chlorophyll content. RGB imagery effectively captures visible traits like size, shape, and greenness but fails to detect early, subtle physiological changes. Stress indicators, such as chlorophyll loss or early water stress, often remain invisible in RGB data. Furthermore, alternative unsupervised models, such as Isolation Forest or Autoencoders, can be evaluated and compared against OCSVM. Also, feature importance analysis or dimensionality reduction techniques could be incorporated to refine the feature space and enhance interpretability. Physiological measurements or field-validated training data may also be introduced to enable supervised or semi-supervised deep learning models for more refined classification and more rigorous validation. Lastly, this pipeline can be generalized and tested on other crops, broadening its applicability in smart agriculture and crop management systems.

## Figures and Tables

**Figure 1 sensors-25-05546-f001:**
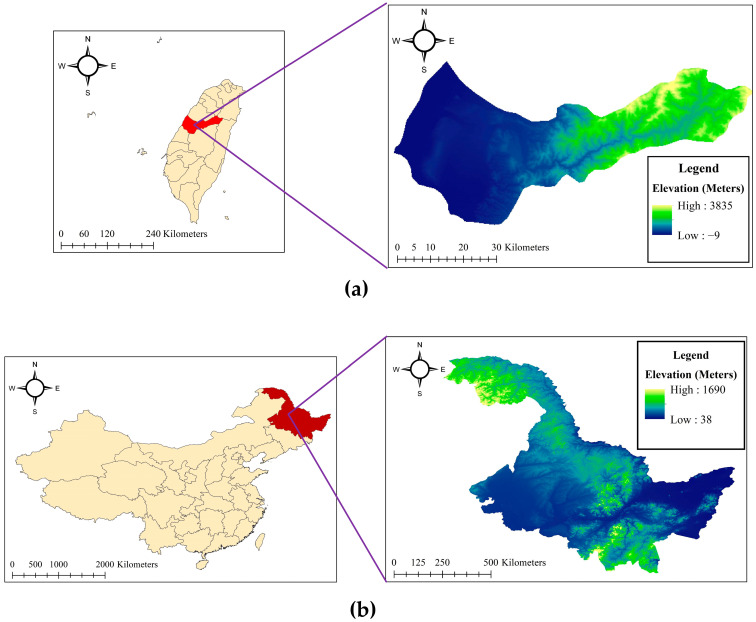
Study area: (**a**) First dataset, located in Taichung province in Taiwan; (**b**) Second dataset, located in Heilongjiang in China.

**Figure 2 sensors-25-05546-f002:**
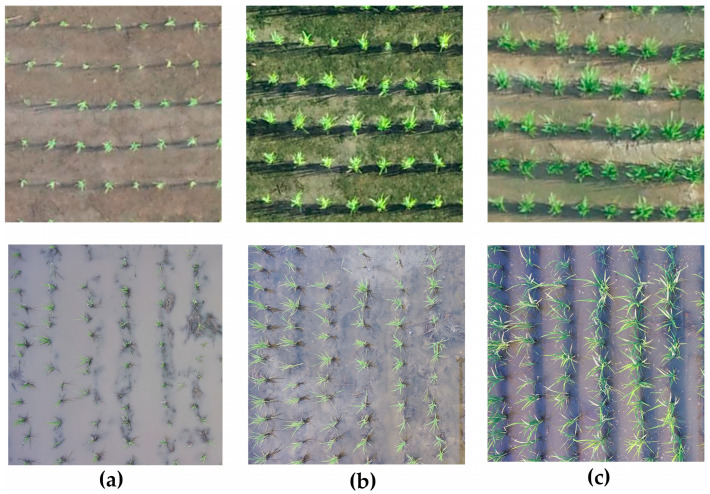
Examples of growth stages of rice seedlings dataset: (**a**) First growth stage; (**b**) Second growth stage; (**c**) Third growth stage. Above and below rows are the images for the first and second datasets, respectively.

**Figure 3 sensors-25-05546-f003:**
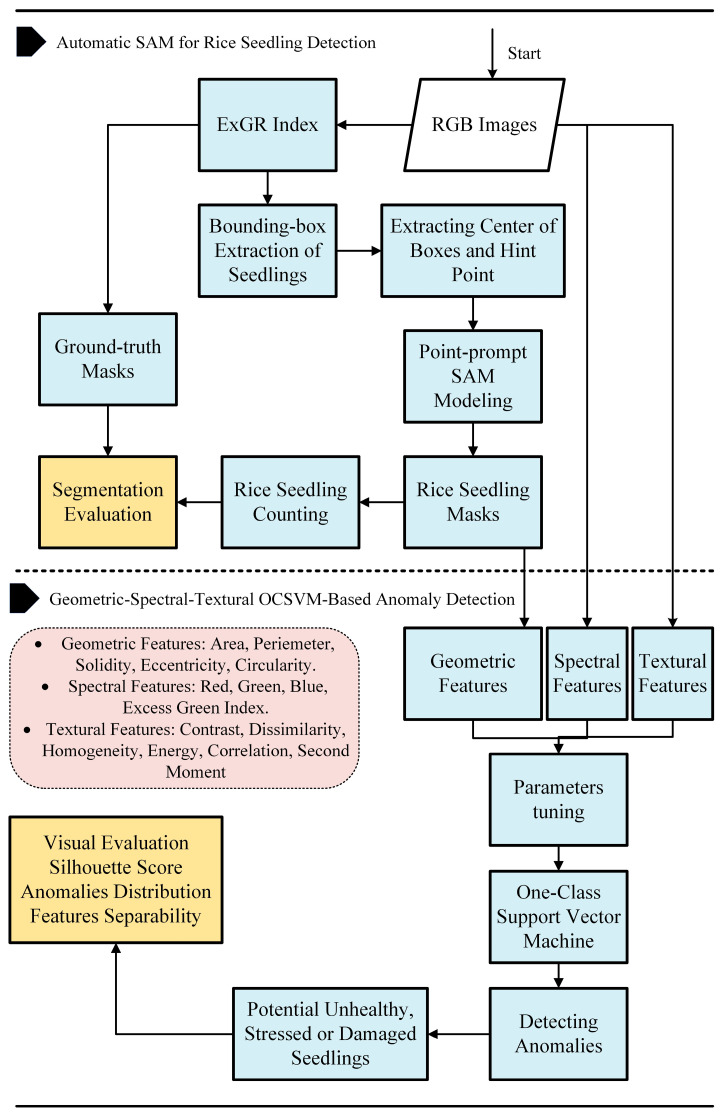
Flowchart of the proposed methodology.

**Figure 4 sensors-25-05546-f004:**
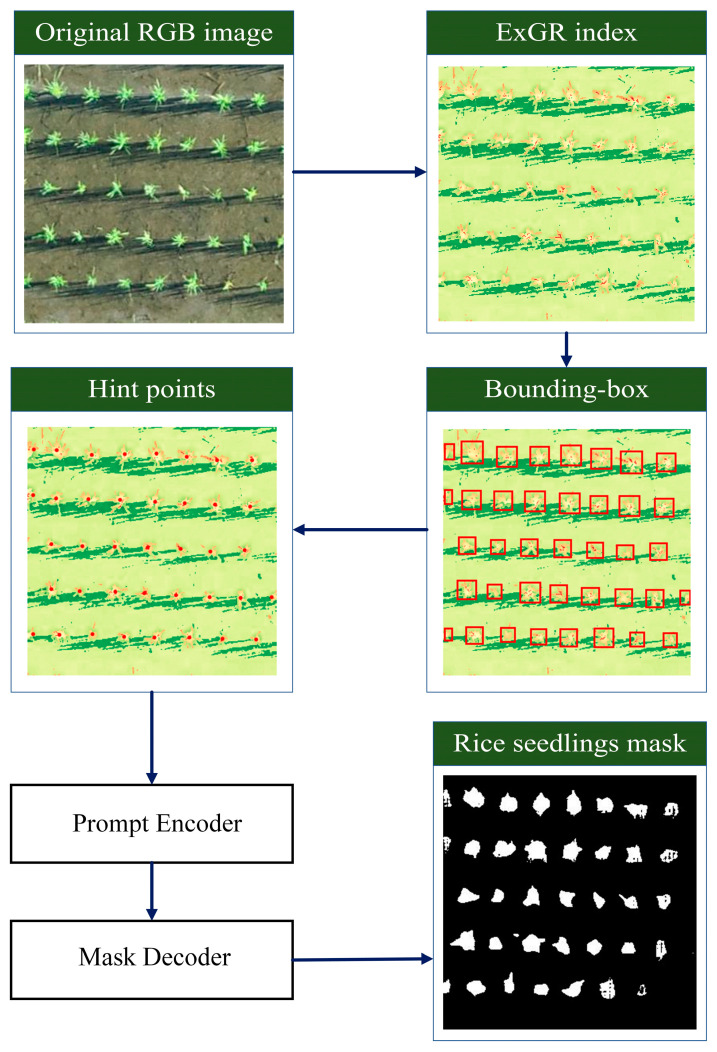
The visual process of Algorithm 1.

**Figure 5 sensors-25-05546-f005:**
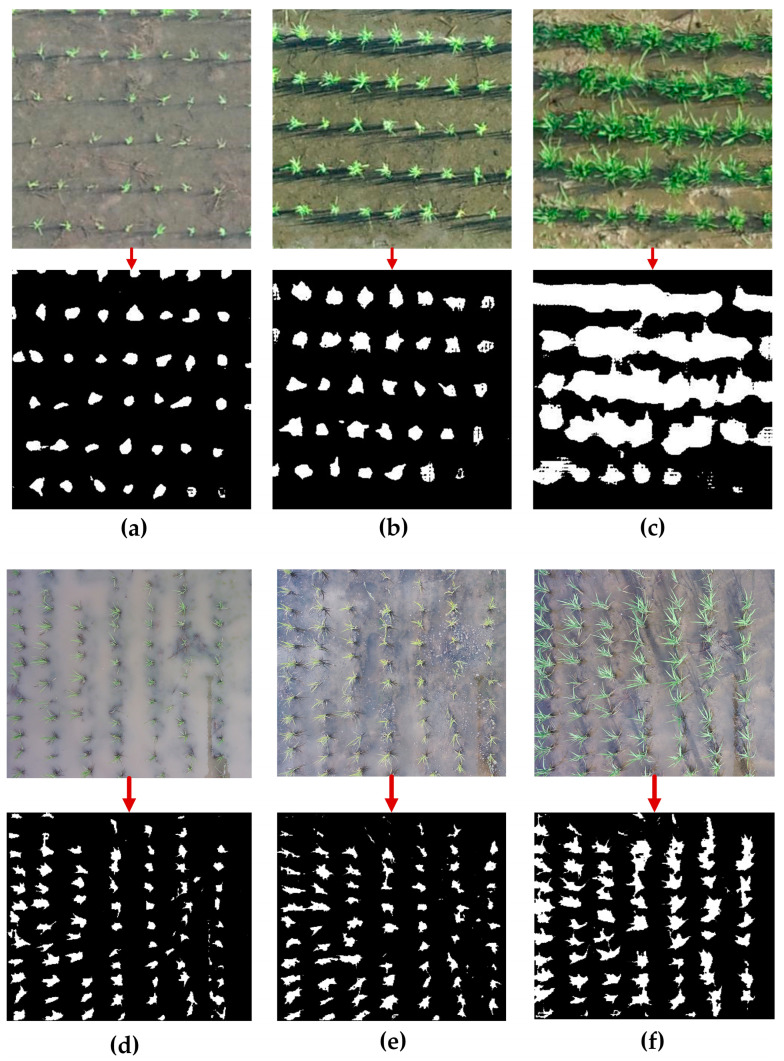
Segmentation results of rice seedlings for varying growth stages: First dataset: (**a**) 7 August 2018; (**b**) 14 August 2018; (**c**) 23 August 2018; Second dataset: (**d**) 1 June 2022; (**e**) 6 June 2022; (**f**) 11 June 2022.

**Figure 6 sensors-25-05546-f006:**
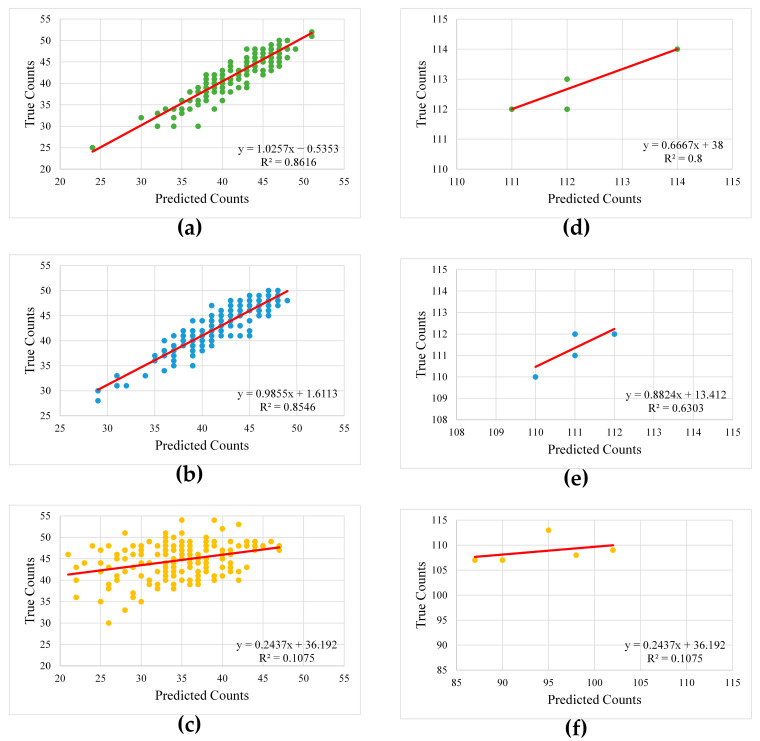
Predicted counts versus the ground truth counts for each growth stage: (**a**–**c**) first, second and third growth stage for the first dataset; (**d**–**f**) first, second and third growth stage for the second dataset, respectively.

**Figure 7 sensors-25-05546-f007:**
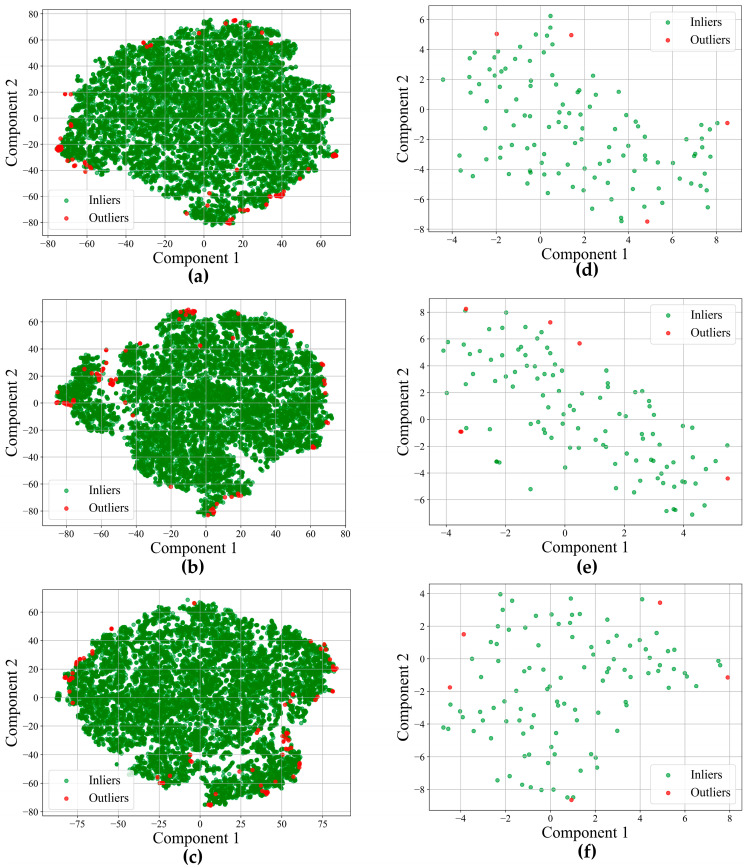
Separability of seedlings in reduced space (2D feature space) obtained from t-SNE, where Component 1 (*x*-axis) and Component 2 (*y*-axis) represent abstract embedding dimensions derived from high-dimensional features: (**a**–**c**) first dataset: 7 August 2018, 14 August 2018, 23 August 2018 and (**d**–**f**) second dataset: 30 May 2022, 7 June 2022, 14 June 2022, respectively.

**Figure 8 sensors-25-05546-f008:**
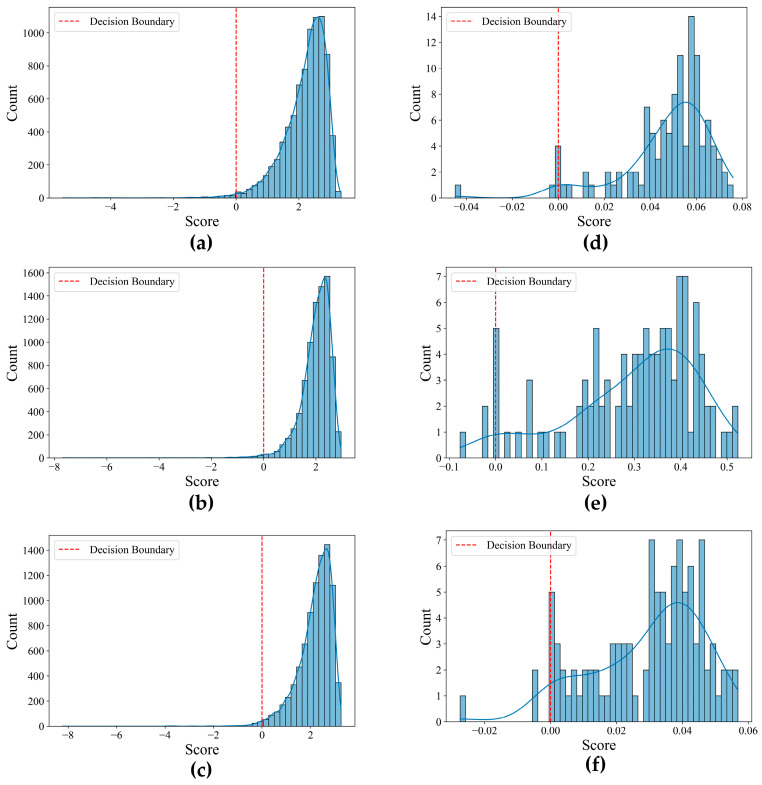
Decision function plot and distance of samples from separating hyperplane: (**a**–**c**) first dataset: 7 August 2018, 14 August 2018, 23 August 2018 and (**d**–**f**) second dataset: 30 May 2022, 7 June 2022, 14 June 2022, respectively.

**Figure 9 sensors-25-05546-f009:**
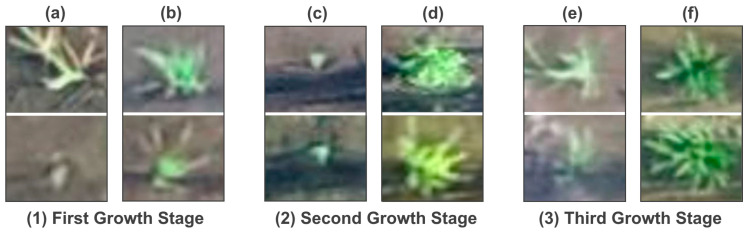
Visual validation of seedlings: (**1**) first growth stage, (**2**) second growth stage, and (**3**) third growth stage. Also, (**a**,**c**,**e**) detected anomalies as damaged or unhealthy seedlings, and (**b**,**d**,**f**) normal seedlings.

**Table 1 sensors-25-05546-t001:** A description of datasets.

Dataset	Spatial Resolution	Temporal Resolution	DOI (Link) of the Dataset
First dataset	2.4 μm	7 days	https://github.com/aipal-nchu/RiceSeedlingDataset (accessed on 30 August 2024)
Second dataset	1 mm	Daily	https://doi.org/10.57760/sciencedb.agriculture.00092 (accessed on 10 June 2025)

**Table 2 sensors-25-05546-t002:** SAM parameters. ViT is short for vision transformer.

Parameter	Value
model_type	ViT_H
checkpoint	sam_vit_h_4b8939.pth
Automatic	False
Predictor	SAM Predictor

**Table 3 sensors-25-05546-t003:** Evaluation of metrics for the proposed automated SAM.

Dataset	Growth Stage	mDice	mIoU	mFPR
First dataset	Early stage (7 August 2018)	94.7	90.1	0.038
	Mid-growth stage (14 August 2018)	91.2	84.0	0.069
	Mature stage (23 August 2018)	72.6	57.7	0.219
Second dataset	Early stage (29 May 2022 to 3 June 2022)	93.0	87.0	0.047
	Mid-growth stage (4–9 June 2022)	85.0	73.8	0.054
	Mature stage (10–14 June 2022)	74.5	59.4	0.075

**Table 4 sensors-25-05546-t004:** Bhattacharyya distance during growth stages.

	First Dataset			Second Dataset		
Features	Stage 1	Stage 2	Stage 3	Stage 1	Stage 2	Stage 3
Area	0.222	0.361	0.326	0.115	0.244	0.222
Perimeter	0.250	0.360	0.215	0.076	0.418	0.230
Solidity	0.084	0.206	0.189	0.170	0.206	0.118
Eccentricity	0.063	0.141	0.152	0.210	0.115	0.067
Circularity	0.488	0.807	0.574	0.092	0.360	0.578
Red	0.193	0.251	0.160	0.275	0.290	0.200
Green	0.169	0.214	0.122	0.314	0.339	0.244
Blue	0.179	0.105	0.205	0.287	0.254	0.377
ExG	0.201	0.068	0.178	0.184	0.197	0.195
Contrast	0.250	0.167	0.218	0.397	0.246	0.320
Dissimilarity	0.264	0.176	0.230	0.339	0.096	0.314
Homogeneity	0.224	0.128	0.196	0.313	0.050	0.147
Energy	0.199	0.305	0.376	0.302	0.129	0.067
Correlation	0.115	0.147	0.148	0.354	0.189	0.177
Second Moment	0.236	0.325	0.925	0.330	0.221	0.099

**Table 5 sensors-25-05546-t005:** Silhouette scores during growth stages.

Dataset	Growth Stage	Silhouette Score
First	Stage 1	0.44
	Stage 2	0.44
	Stage 3	0.41
Second	Stage 1	0.34
	Stage 2	0.31
	Stage 3	0.31

## Data Availability

The first dataset used in this study was obtained from Rice Seedling Dataset repository, originally published in “A UAV Open Dataset of Rice Paddies for Deep Learning Practice” by Yang et al., 2021 [7]. The dataset is publicly available in a GitHub repository and can be accessed at the following link: https://github.com/aipal-nchu/RiceSeedlingDataset (accessed on 30 August 2024). The second dataset was obtained from repository of “Image Dataset of Wheat, Corn, and Rice Seedlings in Heilongjiang Province”, originally published in 2022 by Qin Jia Le and Guo Leifeng [46]. The dataset is publicly available in the Science Data Bank repository and can be accessed at the following link: https://www.scidb.cn/en/detail?dataSetId=a511f28b23444235b5378953c76c47c6#p4 (accessed on 10 June 2025).

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
