# Peer review of "Automated Rice Seedling Segmentation and Unsupervised Health Assessment Using Segment Anything Model with Multi-Modal Feature Analysis"

_sensors, 2025, doi:10.3390/s25175546_

Round 1
Reviewer 1 Report
Comments and Suggestions for Authors
This study holds significant academic value in the field of smart agriculture. It is the first to combine the SAM model with multi-modal features (morphological, spectral, and textural) to achieve fully automated segmentation and unsupervised health assessment of rice seedlings, eliminating the reliance on annotated data required by traditional methods. By generating automated point prompts for SAM using the ExGR index, it addresses the high cost of manual annotation in agricultural scenarios, offering a novel approach for few-shot learning.
- Lack of in-depth adaptation of the SAM model. Only the default parameters (ViT_H) and point prompt function of SAM were used, without optimizing the model architecture or training strategy for agricultural contexts, which may limit performance in complex scenarios (e.g., canopy overlap).
- Morphological/spectral features (e.g., area, RGB channel means) and the ExG index are existing methods, with no introduction of emerging features (e.g., 3D point clouds or chlorophyll fluorescence).
- Dependence on silhouette scores and Bhattacharyya distance without field validation (e.g., photosynthetic rate measurements or pathological tests) to confirm the true stress state of anomalous seedlings.
- Parameters (e.g., ν and γ) were optimized via grid search, but the cross-validation strategy was not specified, risking overfitting.
- Growth stages (e.g., "mature phase") were grouped solely by date, without incorporating phenological indicators (e.g., growing degree days or leaf age), potentially affecting comparability.
- The silhouette scores for the second dataset (0.31–0.34) were significantly lower than those for the first (0.41–0.44), but no explanation was provided (e.g., lighting differences or camera angles).
- Fifteen features (e.g., perimeter, solidity, contrast) were used simultaneously without importance analysis or dimensionality reduction (e.g., PCA), possibly introducing noise.
- The discussion noted "strong discriminative power of circularity in later stages" but failed to contrast it with earlier results (e.g., textural features being more effective).
- The suggestion to "incorporate red-edge bands" lacked justification for why RGB data alone is insufficient.
- "ExGR" in the abstract vs. "ExGR index" in the methods section, while Formula (1) uses "ExGR." Recommend standardizing the abbreviation.
- "mDice" and "Dice coefficient" were used interchangeably (Page 7 and Table 3). Recommend unifying to "mDice" (mean form).
Author Response
Reviewer 1:
Comments and Suggestions for Authors
This study holds significant academic value in the field of smart agriculture. It is the first to combine the SAM model with multi-modal features (morphological, spectral, and textural) to achieve fully automated segmentation and unsupervised health assessment of rice seedlings, eliminating the reliance on annotated data required by traditional methods. By generating automated point prompts for SAM using the ExGR index, it addresses the high cost of manual annotation in agricultural scenarios, offering a novel approach for few-shot learning.
Dear reviewer,
We would like to thank you very much for your time, efforts, and constructive comments that helped us improved the presentation of our manuscript. We have tried to address them all. Please see below our point-by-point responses, where the changes are highlighted in the revised manuscript.
Lack of in-depth adaptation of the SAM model. Only the default parameters (ViT_H) and point prompt function of SAM were used, without optimizing the model architecture or training strategy for agricultural contexts, which may limit performance in complex scenarios (e.g., canopy overlap).
- Authors’ Response: We appreciate the reviewer’s valuable observation. Our intention in this study was to evaluate SAM’s baseline performance under minimal adaptation, in order to test the generalization capacity of foundation models in agricultural imagery and establish a reference point for rice seedling detection. To this end, we employed the ViT-H backbone with point prompts and further automated the process of placing hint points to compensate for the absence of field annotations, aligning SAM more closely with agricultural image characteristics. While we acknowledge that deeper adaptations or fine-tuning could further enhance performance in complex scenarios, such work lies beyond the scope of this paper and will be addressed in limitations. We have clarified this motivation in the revised manuscript.
- Morphological/spectral features (e.g., area, RGB channel means) and the ExG index are existing methods, with no introduction of emerging features (e.g., 3D point clouds or chlorophyll fluorescence).
- Authors’ Response: In the section regarding future research ideas, it is mentioned that the framework can be integrated with more modalities, such as NIR and red-edge, thermal bands, 3D point clouds or chlorophyll fluorescence to extract physiological and structural indicators.
- Dependence on silhouette scores and Bhattacharyya distance without field validation (e.g., photosynthetic rate measurements or pathological tests) to confirm the true stress state of anomalous seedlings.
- Authors’ Response: Thank you for your insightful comment. Since both datasets are benchmark datasets for years 2018 and 2022, field or physiological measurements are not available. For this reason, it is suggested that in future research, physiological data and measurements of plant stress parameters should be collected during field visits.
- Parameters (e.g., ν and γ) were optimized via grid search, but the cross-validation strategy was not specified, risking overfitting.
- Authors’ Response: Thank you for your insightful comment. To mitigate the risk of overfitting during parameter optimization, we employed the silhouette score as a validation metric within the grid search process. This allowed us to evaluate the quality of separation between the identified clusters without relying on labeled data. Specifically, ν and γ were tuned by selecting the parameter set that maximized the average silhouette score, ensuring that the chosen model favored well-structured separation between normal and anomalous samples. While we acknowledge that unsupervised validation lacks the full rigor of supervised cross-validation, this approach provided a principled and robust way to guide parameter selection in the absence of ground-truth labels.
- Growth stages (e.g., "mature phase") were grouped solely by date, without incorporating phenological indicators (e.g., growing degree days or leaf age), potentially affecting comparability.
- Authors’ Response: Thank you for your insightful comment. UAV acquisitions were scheduled at fixed calendar dates to represent distinct growth stages, a practical compromise in the absence of detailed phenological measurements, and an approach commonly used in UAV-based monitoring studies. Therefore, in this study, growth stages were grouped by acquisition date rather than phenological indicators, which may limit the biological comparability of the stages. As we mentioned in future studies section, incorporating field-based measurements can enhance biological consistency in stage comparisons.
- The silhouette scores for the second dataset (0.31–0.34) were significantly lower than those for the first (0.41–0.44), but no explanation was provided (e.g., lighting differences or camera angles).
- Authors’ Response: Thank you for your insightful comment. Despite these lower values, the silhouette scores remain within an acceptable range for unsupervised clustering tasks under highly variable outdoor field conditions. These lower scores can be attributed to more variable illumination, diverse backgrounds (soil and flooded), or potential differences in UAV imaging distance, which may reduce the separability of feature distributions. The statement is added to the text for clarification.
- Fifteen features (e.g., perimeter, solidity, contrast) were used simultaneously without importance analysis or dimensionality reduction (e.g., PCA), possibly introducing noise.
- Authors’ Response: Thank you for your insightful comment. We appreciate the comment on the simultaneous use of fifteen features. Our decision to retain all features was motivated by the unsupervised nature of OCSVM, where discarding features without ground-truth labels may risk losing information relevant to anomaly detection. The selected spectral and geometrical features were carefully chosen to represent complementary aspects of seedling health and morphology. Furthermore, the stability of the silhouette score across parameter settings suggests that the model was not overly sensitive to potential feature noise. Nonetheless, we agree that feature importance analysis or dimensionality reduction (e.g., PCA) could further refine the feature space and enhance model interpretability. We plan to integrate such approaches in future work to better understand the contribution of each feature.
- The discussion noted "strong discriminative power of circularity in later stages" but failed to contrast it with earlier results (e.g., textural features being more effective).
- Authors’ Response: Thank you for your insightful comment. A clarification statement is added to the related section to highlight how the importance shifts across growth stages. The added statement: “In contrast, during the first growth stage, textural attributes such as contrast and dissimilarity, together with some spectral features (e.g., R and G channels), exhibited higher discriminative power, reflecting the stronger color and texture differences between seedlings and background at early stages. Interestingly, in the third stage of the first dataset, the second moment also reached its maximum discriminative value, further supporting the stage-dependent feature shift.”
- The suggestion to "incorporate red-edge bands" lacked justification for why RGB data alone is insufficient.
- Authors’ Response: Thank you for your insightful comment. RGB imagery is effective for detecting visual traits, such as seedling size, shape, and basic greenness, but it has limitations in capturing subtle physiological changes before they are visible. For example, stress symptoms, such as reduced chlorophyll content or early water stress often occur at the cellular level and are not fully reflected in RGB color values. Red-edge and NIR bands are more sensitive to vegetation biochemistry and canopy structure. Incorporating these bands would therefore enhance the framework’s ability to capture early stress indicators and improve robustness under variable environmental conditions where RGB reflectance alone may be insufficient.
Comments on the Quality of English Language
- "ExGR" in the abstract vs. "ExGR index" in the methods section, while Formula (1) uses "ExGR." Recommend standardizing the abbreviation.
- Corrected. Thank you!
- "mDice" and "Dice coefficient" were used interchangeably (Page 7 and Table 3). Recommend unifying to "mDice" (mean form).
- All are unified. Thank you!
We hope the changes that we have made are satisfactory.
Thank you again for your time and insightful comments
Sincerely,
Authors
Reviewer 2 Report
Comments and Suggestions for Authors
First of all, I do not have experience in the Health or Agriculture industry, so my review resumes strictly from the technical point of view.
Strengths
-
Well-structured introduction with strong motivation based on global food demand.
-
Rich reference list and clear identification of gaps in plant monitoring and remote sensing.
-
Novel integration of ExGR index with the Segment Anything Model (SAM) for automated segmentation.
-
Use of OCSVM for anomaly detection is appropriate, avoiding the need for large labeled datasets.
-
Demonstrates applicability across three rice growth stages, supporting temporal monitoring.
Weaknesses / Concerns
-
The term “fully automated” is not clearly defined or justified.
- Validation relies heavily on visual inspection; no physiological or ground-truth health data provided, and no actual comparison with the real data from 2018 and 2022.
-
Limited to RGB imagery; performance may be constrained compared to multispectral or hyperspectral approaches.
-
Computational efficiency and robustness under field conditions are not discussed.
Suggestions for Improvement
-
Clarify what is meant by “fully automated” and whether any manual steps are still required.
- Consider integrating additional sensing modalities to capture subtle stress indicators.
-
Provide more rigorous validation beyond visual inspection, ideally with field or physiological measurements.
-
Discuss scalability and real-world applicability, especially in variable outdoor environments.
Overall Evaluation
The paper presents an interesting and technically relevant approach to crop monitoring by combining segmentation and anomaly detection. While the segmentation results are promising, the anomaly detection component and validation could be strengthened to better support the claims of a “fully automated” health assessment framework.
Author Response
Reviewer 2:
Comments and Suggestions for Authors
First of all, I do not have experience in the Health or Agriculture industry, so my review resumes strictly from the technical point of view.
Dear reviewer,
We would like to thank you very much for your time, efforts, and constructive comments that helped us improved the presentation of our manuscript. We have tried to address them all. Please see below our point-by-point responses, where the changes are highlighted in the revised manuscript.
Strengths
Well-structured introduction with strong motivation based on global food demand.
Rich reference list and clear identification of gaps in plant monitoring and remote sensing.
Novel integration of ExGR index with the Segment Anything Model (SAM) for automated segmentation.
Use of OCSVM for anomaly detection is appropriate, avoiding the need for large labeled datasets.
Demonstrates applicability across three rice growth stages, supporting temporal monitoring.
Weaknesses / Concerns
The term “fully automated” is not clearly defined or justified.
Validation relies heavily on visual inspection; no physiological or ground-truth health data provided, and no actual comparison with the real data from 2018 and 2022.
Limited to RGB imagery; performance may be constrained compared to multispectral or hyperspectral approaches.
Computational efficiency and robustness under field conditions are not discussed.
Suggestions for Improvement
Clarify what is meant by “fully automated” and whether any manual steps are still required.
- Authors’ Response: Thank you for your insightful comment. By “fully automated,” we mean that once the framework is provided with UAV imagery, all subsequent processing steps, including vegetation enhancement (ExGR with multi-Otsu thresholding), feature extraction, parameter selection (via grid search for OCSVM), and anomaly detection are executed automatically without the need for manual intervention. These processes are algorithmically defined and do not require subjective user tuning or manual adjustment during execution. In this sense, the entire pipeline operates in a fully automated manner after initialization. We have clarified this in the revised manuscript. These are now mentioned and highlighted in the Discussion section.
Consider integrating additional sensing modalities to capture subtle stress indicators.
- Authors’ Response: Thank you for your insightful comment. In the section regarding future research ideas, it is mentioned that the framework can be integrated with more modalities, such as NIR and red-edge, thermal bands, 3D point clouds or chlorophyll fluorescence to extract physiological and structural indicators.
Provide more rigorous validation beyond visual inspection, ideally with field or physiological measurements.
- Authors’ Response: Thank you for your insightful comment. Since both datasets are benchmark datasets for years 2018 and 2022, field or physiological measurements are not available. For this reason, it is suggested that in future research, physiological data and measurements of plant stress parameters should be collected during field visits.
Discuss scalability and real-world applicability, especially in variable outdoor environments.
- Authors’ Response: Thank you for your insightful comment. There are four important points: 1) The unsupervised design of our anomaly detection approach makes it suitable for large-scale monitoring without requiring extensive labeled datasets; 2) The zero-shot approach of SAM allows scalable applicability and can be applied in a patch-wise manner, results in efficient processing of large UAV datasets and enables extension to broader spatial scales without tuning according to a dataset; 3) Our methodology was tested on two UAV datasets with distinct soil and flooded background conditions as well as under different illumination scenarios, shows its robustness across variable environmental and 4) It is acknowledged that illumination variability, soil background heterogeneity, and crop phenological differences can influence segmentation and anomaly detection performance. A paragraph according to these four points is added in the discussion about scalability.
Overall Evaluation
The paper presents an interesting and technically relevant approach to crop monitoring by combining segmentation and anomaly detection. While the segmentation results are promising, the anomaly detection component and validation could be strengthened to better support the claims of a “fully automated” health assessment framework.
We hope the changes that we have made are satisfactory.
Thank you again for your time and insightful comments
Sincerely,
Authors
Reviewer 3 Report
Comments and Suggestions for Authors
The authors propose a two-step framework for automated rice seedling segmentation and health assessment, integrating spectral, morphological, and textural features to enhance agricultural monitoring and evaluation. The method is validated across three stages of rice growth and shows potential for agricultural applications.
The language is proficient and conveys the information effectively, although some domain-specific terms (e.g., multi-modal features, spectral signature) could be briefly defined for broader readability.
ABOUT THE EXPOSITION
The exposition of the paper is generally well-structured and provides a coherent overview of the proposed method for automated rice seedling segmentation and unsupervised health assessment. A detailed methodology flowchart is helpful to visualize the key stages of the proposed method.
The discussion section is extensive as it should be; however, some factual information could be presented earlier in the paper. For example, additional descriptions of solidity, eccentricity, circularity, and ExG attributes should belong to Section 2.2.2. Also, information on how to distinguish unhealthy seedlings (Lines 367-369) should not be posted in the discussion, but much earlier.
IMPORTANT
- The paper sometimes uses „morphological features“ and sometimes „geometric attributes“ or „geometric features“. From the context, it appears that the geometric attributes (area, perimeter, solidity, eccentricity, circularity) are the specific measures used to represent morphological features. It would be clearer for readers if the authors explained what features are and what attributes are.
- Silhouette scores, ranging from 0.31 to 0.44, for health assessment appear relatively low, suggesting that effective anomaly separation may not be claimed. Furthermore, the use of „visual inspection“ as a validation method lacks specificity; i.e., details on how it was conducted, how it was measured (quantitative score), and its impact on visual validation, together with SH, are not provided.
- Please define (5), (6), and (7) formulas unambiguously. The meaning of Area, Convex area, Minor axis/Major axis, and perimeter need to be explained in the context of this study. Also, the manuscript currently lists circularity as C = 4π/P, which appears to be incomplete. In image analysis, circularity is commonly defined as 4πA/P^2. (A – area and P – perimeter). Additionally, if the authors used MATLAB’s regionprops function, note that MATLAB applies a correction to compensate for pixel-based perimeter measurement in that case: C = (4πA/P^2) * (1 − 0.5/r)^2, where r = P/(2π) + 0.5. Clarify definition and revise the formula.
- The article does not specify how texture features are calculated, what methods are used, etc.
MINOR REMARKS
- In the Abstract (Line 19), the term „Excess Green Red Minus Excess Red Index (ExGR)“ appears to be a typo. It should be corrected to „Excess Green Minus Excess Red Index (ExGR)“ to align with standard terminology. Note that the Abbreviations list in line 440 instead contains the correct definition compared to the Abstract.
- The definitions of true positive (TP), false positive (FP) and false negative (FN) should be added after (2), (3), and (4) formulas in the main text and/or included (spelt out in full) in the abbreviations list.
- Silhouette is the only measure in the paper that does not have a mathematical definition (or citation). As a reminder, this measure also has a nice graphical representation (in Table 5). Similarly, Bhattacharyya distance can also be defined mathematically for clarity, or a citation can be given.
- Considering visual inspection is one of the measures for better visual presentation (and identification of anomalies), the seedlings in Figure 9 could benefit from higher resolution (less blurry) images.
- Description of Figure 7 could explain (remind) Component 1 and 2 (x, y) label meanings.
- In Line 78, One-Class Support Vector Machine (OCSVM) was already spelt out in full in the Abstract. In Line 68, The Segment Anything Model (SAM) was already spelt out in full in the Abstract.
- Compared to other features, textural features do not seem to be included (represented) in the Flowchart of the proposed methodology (Figure 3). Moreover, the starting point of the Flowchart could be clarified (notation) and the colors (of the boxes) explained if important.
- The Abstract (Line 14) describes the framework as „fully automated“, but it is unclear whether some manual steps, such as parameter tuning for ExGR or OCSVM, are involved/required. The authors should clarify the extent of automation in the conclusions to support this claim.
Author Response
Reviewer 3:
Comments and Suggestions for Authors
The authors propose a two-step framework for automated rice seedling segmentation and health assessment, integrating spectral, morphological, and textural features to enhance agricultural monitoring and evaluation. The method is validated across three stages of rice growth and shows potential for agricultural applications.
The language is proficient and conveys the information effectively, although some domain-specific terms (e.g., multi-modal features, spectral signature) could be briefly defined for broader readability.
Dear reviewer,
We would like to thank you very much for your time, efforts, and constructive comments that helped us improved the presentation of our manuscript. We have tried to address them all. Please see below our point-by-point responses, where the changes are highlighted in the revised manuscript.
ABOUT THE EXPOSITION
The exposition of the paper is generally well-structured and provides a coherent overview of the proposed method for automated rice seedling segmentation and unsupervised health assessment. A detailed methodology flowchart is helpful to visualize the key stages of the proposed method.
The discussion section is extensive as it should be; however, some factual information could be presented earlier in the paper. For example, additional descriptions of solidity, eccentricity, circularity, and ExG attributes should belong to Section 2.2.2. Also, information on how to distinguish unhealthy seedlings (Lines 367-369) should not be posted in the discussion, but much earlier.
Authors’ Response: Thank you for your insightful comments. We have revised the manuscript according to your suggestions.
IMPORTANT
The paper sometimes uses “morphological features” and sometimes “geometric attributes“ or “geometric features”. From the context, it appears that the geometric attributes (area, perimeter, solidity, eccentricity, circularity) are the specific measures used to represent morphological features. It would be clearer for readers if the authors explained what features are and what attributes are.
- Authors’ Response: Thank you for your insightful comment. “Attributes” is used as a synonym for “features”. They are replaced with “feature” now.
Silhouette scores, ranging from 0.31 to 0.44, for health assessment appear relatively low, suggesting that effective anomaly separation may not be claimed. Furthermore, the use of “visual inspection” as a validation method lacks specificity; i.e., details on how it was conducted, how it was measured (quantitative score), and its impact on visual validation, together with SH, are not provided.
- Authors’ Response: Thank you for your insightful comment. Despite these lower values, the silhouette scores remain within an acceptable range for unsupervised clustering tasks under highly variable outdoor field conditions. As explained in the text, visual inspection is one part of a multi-pronged validation strategy. The paragraph now is more refined to be clear about how it was conducted. Please see Section 2.2.2.
“To assess the performance of the proposed approach, several validation strategies were applied. First, anomaly seedlings were overlaid on UAV imagery to visualize the spatial distribution of detected outliers, allowing direct inspection of their physical characteristics. This qualitative step was carried out by the research team to verify whether detected anomalies corresponded to visibly distinct seedlings. Additionally, the Bhattacharyya distance was calculated to quantify the degree of overlap between clusters, with higher values indicating better separability”
Please define (5), (6), and (7) formulas unambiguously. The meaning of Area, Convex area, Minor axis/Major axis, and perimeter need to be explained in the context of this study. Also, the manuscript currently lists circularity as C = 4π/P, which appears to be incomplete. In image analysis, circularity is commonly defined as 4πA/P^2. (A – area and P – perimeter). Additionally, if the authors used MATLAB’s regionprops function, note that MATLAB applies a correction to compensate for pixel-based perimeter measurement in that case: C = (4πA/P^2) * (1 − 0.5/r)^2, where r = P/(2π) + 0.5. Clarify definition and revise the formula.
- Authors’ Response: Thank you for your insightful comment. The definition in the context of this study is added to the text: “Area is the number of pixels inside the seedling mask, Convex_area is the smallest convex polygon that encloses the seedling mask, Perimeter is the length of the boundary of the seedling mask, and Major_ and Minor_axis lengths are the length of the longest and shortest axis of the best-fitting ellipse to the seedling shape.”. Yes, “Area” was missing in the formulae; It is corrected. The circularity is computed using handwritten formulae in python as below:
circularity = (4 * numpy.pi * area) / (perimeter ** 2)
The article does not specify how texture features are calculated, what methods are used, etc. Textural features are extracted from GLCM matrix using Scikit-image library.
MINOR REMARKS
In the Abstract (Line 19), the term „Excess Green Red Minus Excess Red Index (ExGR)“ appears to be a typo. It should be corrected to „Excess Green Minus Excess Red Index (ExGR)“ to align with standard terminology. Note that the Abbreviations list in line 440 instead contains the correct definition compared to the Abstract.
- Authors’ Response: Thank you for your insightful comment. Corrected.
The definitions of true positive (TP), false positive (FP) and false negative (FN) should be added after (2), (3), and (4) formulas in the main text and/or included (spelt out in full) in the abbreviations list.
- Authors’ Response: Thank you for your insightful comment. The definitions are added both in the main text and in the abbreviations list. TP (True Positives) are pixels correctly identified as seedlings, while TN (True Negatives) are pixels correctly identified as background. FP (False Positives) occur when background pixels are incorrectly labeled as seedlings, and FN (False Negatives) occur when actual seedling pixels are missed and labeled as background.
Silhouette is the only measure in the paper that does not have a mathematical definition (or citation). As a reminder, this measure also has a nice graphical representation (in Table 5). Similarly, Bhattacharyya distance can also be defined mathematically for clarity, or a citation can be given.
- Authors’ Response: Thank you for your insightful comment. Citations are added. References [47], [48], [49].
Considering visual inspection is one of the measures for better visual presentation (and identification of anomalies), the seedlings in Figure 9 could benefit from higher resolution (less blurry) images.
- Authors’ Response: Thank you for your insightful comment. We improved the quality of this image.
Description of Figure 7 could explain (remind) Component 1 and 2 (x, y) label meanings.
- Authors’ Response: Thank you for your insightful comment. Description is added: “Component 1 (x-axis) and Component 2 (y-axis) represent abstract embedding dimensions derived from high-dimensional features”
In Line 78, One-Class Support Vector Machine (OCSVM) was already spelt out in full in the Abstract. In Line 68, The Segment Anything Model (SAM) was already spelt out in full in the Abstract.
- Authors’ Response: Thank you for your insightful comment. We now define these acronyms in the manuscript not in the abstract.
Compared to other features, textural features do not seem to be included (represented) in the Flowchart of the proposed methodology (Figure 3). Moreover, the starting point of the Flowchart could be clarified (notation) and the colors (of the boxes) explained if important.
- Authors’ Response: Thank you for your insightful comment. Flowchart is corrected.
The Abstract (Line 14) describes the framework as “fully automated”, but it is unclear whether some manual steps, such as parameter tuning for ExGR or OCSVM, are involved/required. The authors should clarify the extent of automation in the conclusions to support this claim.
- Authors’ Response: Thank you for your insightful comment. The framework is fully automated in the sense that both seedling extraction and classification rely on algorithmic steps without manual intervention: ExGR thresholding is determined automatically via multi-Otsu (recently added to the text), and OCSVM hyperparameters are selected through automated grid search (automated parameter selection). No manual parameter tuning is required once the pipeline is executed. We highlighted these in the discussion section.
We hope the changes that we have made are satisfactory.
Thank you again for your time and insightful comments
Sincerely,
Authors